# Efficient Attention via Pre-Scoring: Prioritizing Informative Keys in Transformers

## Abstract

Recent advances in transformer architectures deeply enhanced long-context language modeling. Among them, HyperAttention achieves competitive efficiency by combining a single-level LSH-based clustering with uniform residual sampling. However, HyperAttention fails to find all significant keys, which in turn raises the overall perplexity. We propose a pre-scoring mechanism that prioritizes significant keys before applying HyperAttention. We introduce three scoring methods: $k$-means and kernel $k$-means clustering, $k$-median clustering, and leverage score-based ranking (inspired by LevAttention) to filter keys effectively. We further replace HyperAttention's original uniform residual sampling, relying exclusively on our pre-scoring mechanism. Experiments on ChatGLM2 (131k token context) reduce perplexity from 12 to 8.3, which outperforms standard HyperAttention. Moreover, when running on the Vision-Transformer (ViT), our method shows that it can guarantee similar accuracy compared with LevAttention, and will surpass LevAttention given specific parameters. Although this method introduces some computational overhead, its combination with HyperAttention achieves up to 20 times faster than FlashAttention, providing a balanced trade-off between speed and modeling accuracy. Our results highlight the effectiveness of integrating pre-scoring into hierarchical attention mechanisms, significantly improving transformer efficiency.

## 1 Introduction

Transformer-based large language models (LLMs) now underpin cutting-edge performance across diverse applications, including computer vision Bi et al. (2021) and text classification Rodrawangpai & Daungjaiboon (2022), yet their quadratic self-attention cost remains a persistent barrier to efficient long-context processing. Especially for Transformer-based attention mechanisms, the time complexity rises quadratically with respect to the sequence length. Without effective strategies to alleviate this cost, deploying transformer models on tasks requiring truly long contexts remains impractical and challenging.

Consider the input of each attention layer, which is typically represented as an $n \times d$ matrix $X$, where $n$ denotes the context length and $d$ is the embedding dimension of the tokens. From this input matrix, we generate three distinct matrices by applying learned linear transformations. Specifically, we compute the query matrix $Q = X \cdot W_Q$, the key matrix $K = X \cdot W_K$, and the value matrix $V = X \cdot W_V$, where $W_Q$, $W_K$, and $W_V$ are learned parameter matrices of dimensions $d \times d$. Various methods have been developed to improve efficiency when processing $Q$, $K$, and $V$. For example, Performer Choromanski et al. (2022) replaces the standard softmax attention with kernel methods. This remarkably reduces computational complexity, yet it involves potential kernel approximation-induced errors. FlashAttention Dao et al. (2022) facilitates computation by optimizing memory access patterns during the matrix computations. Despite its advantages, specialized hardware support is required for optimal performance.

More recent advances in efficient attention methods have explored diverse strategies to mitigate quadratic complexity while maintaining performance. For long-context processing, DuoAttention Xiao et al. (2024) combines retrieval-augmented heads with streaming attention mechanisms, dynamically balancing local and global context access. The adaptive approach in "Unveiling Simplicities

of Attention" Donhauser et al. (2025) identifies and preserves only essential attention heads for long-context modeling, achieving comparable performance with significantly reduced computation.

In addition, HyperAttention Han et al. (2023) reduces the computational burden using approximate matrix product and locality sensitive hashing. These methods improve efficiency at the cost of some degradation in perplexity. LevAttention Kannan et al. (2024) selects a fixed set of keys independent of the queries to try to capture all heavy attention scores. However, it is less effective in capturing query-specific attention patterns.

Inspired by HyperAttention and LevAttention, we propose a pre-scoring mechanism. Namely, we use clustering-based filtering to prioritize informative keys before applying HyperAttention.

**Our Results** We first show that clustering-based pre-scoring *consistently outperforms* leverage-score selection: for example, on a Vision Transformer (ViT) Large model Dosovitskiy et al. (2021) (85.85% baseline accuracy on ImageNet-1k), sampling 128 keys via K-means retains **84.46%** accuracy, compared to only **77.17%** when using leverage scores (top-128). Furthermore, by integrating our clustering-based pre-scoring into HyperAttention, we reduce *ChatGLM2-6b-32k*'s perplexity on the *Longbench* data set Bai et al. (2024) from 12 to 8.3, outperforming both the original HyperAttention and the variant augmented with leverage-score selection (LevAttention+HyperAttention) GLM et al. (2024); Kannan et al. (2024). Moreover, our method preserves the computational efficiency advantages of HyperAttention over standard self-attention and FlashAttention Dao et al. (2022).

In addition to our empirical results, we revisit the planted-subspace model introduced for LevAttention (Section 4 of LevAttentionKannan et al. (2024)), and we show that our pre-scoring clustering-based methods recover the same planted model guarantees of LevAttention, in terms of recovering all heavy keys for a query by a Markov bound with constant probability $1 - 1/\mathrm{poly}(d)$. Thus, theoretically our methods are no worse than those of LevAttention in this natural planted model. We further introduce an alternative planted-subspace model in which K-means clustering provably recovers keys with large leverage score, showing again that K-means pre-scoring is as powerful as leverage score pre-scoring for natural models. This may help explain why our empirical results consistently outperform those of LevAttention Kannan et al. (2024).

## 2 PRELIMINARIES

We define the attention mechanism $Att = D^{-1}AV$. Here, $A$ is defined as $A := \exp(QK^\top)$, while $D$ is a diagonal matrix with $D_{i,i} = \|A_{i,:}\|_1$ for each $i \in [n] = \{1, 2, \dots, n\}$. We refer to the matrix $A$ as the attention matrix. Explicitly calculating the attention matrix requires $\Theta(n^2 d)$ time and $\Theta(n^2)$ memory, which can be prohibitive.

Han et al. introduce HyperAttention Han et al. (2023), which addresses the quadratic bottleneck of vanilla self-attention by hashing queries and keys with an angular locality-sensitive hash (LSH) function and then ordering buckets so that adjacent buckets have small Hamming distance. Scores are computed only for pairs that fall into the same hash bucket, significantly reducing the time and memory cost on typical data distributions. Their method performs an additional randomized low-rank compression inside each bucket, further reducing constant factors. Its main limitation is data independence: because bucket membership is determined solely by the hash function, weak but semantically crucial long-range links may never collide and can therefore be missed.

LevAttention Kannan et al. (2024) takes a complementary view. It first sketches the key matrix in $O(nd)$ time to approximate statistical leverage scores for each row of the key matrix, then forms a Universal set $U = \{i : \mathrm{LS}(K_i) \geq \epsilon\}$ that, for polynomial-based attention (rather than the standard softmax attention), is guaranteed to contain all attention scores whose weight exceeds a user-chosen threshold $\epsilon > 0$. Looking only at $U$ therefore achieves perfect recall of heavy attention scores, independent of positional locality. However, when $\epsilon$ is very small, $|U|$ can be large as $n$, eliminating any savings.

Moreover, a uniform evaluation within the set $U$ wastes time in many pairs with low impact. By first constructing $U$ to ensure theoretical coverage and then applying HyperAttention's locality-sensitive hashing inside this recall-guaranteed set, our hybrid approach combines the strengths of both methods: it retains subquadratic complexity and provably captures every $\epsilon$-heavy attention.

Our goal is to accelerate approximation methods for transformers while maintaining high accuracy. Rather than using the universal set $U$ of Kannan et al. Kannan et al. (2024), we suggest other sets based on clustering methods to guide approximation algorithms such as HyperAttention Han et al. (2023) to find large attention scores.

# 3 ALGORITHM

A central motivation for our pre-scoring approach is the hypothesis that computationally efficient methods, such as clustering, can effectively identify and prioritize the most salient keys within the key matrix $K \in \mathbb{R}^{n \times d_k}$ (where $n$ is sequence length and $d_k$ is the key dimension). This idea is supported by Axiotis et al. Axiotis et al. (2024). They developed a cluster-based sensitivity sampling method to enhance data selection efficiency for large-scale model training. This principle resonates with techniques such as LevAttention Kannan et al. (2024), which use statistical leverage scores to sample influential data points. To provide rigorous grounding for clustering, and specifically K-means or K-median) as a pre-scoring mechanism, we analyze its performance and relationship to leverage scores under a structured data model.

## 3.1 STRUCTURAL GUARANTEES

### 3.1.1 MATRIX STRUCTURE AND ASSUMPTIONS

*The guarantees in this section follow Kannan et al. (2024) and apply to polynomial attention; softmax results are empirical. Unless stated otherwise:*

**Assumption 1** (Model and regularity). *Keys split into $S \cup N$ with means $\mu_S, \mu_N$, within-cluster variance $\leq \sigma^2$, separation $\Delta = \|\mu_S - \mu_N\|_2$; analysis uses a degree-$r$ polynomial kernel; the selector retains $s$ keys per query; all probability statements are over the data and algorithmic randomness.*

We give a new planted model for which cluster-based prescoring recovers all large leverage scores.

Let $A \in \mathbb{R}^{n \times d}$ be a matrix generated as follows:

1. There are $d$ disjoint sets of row indices, $S_1, \ldots, S_d$, each of size $m = \lceil 1/\epsilon \rceil$ for some small $\epsilon \in (0, 1)$.

2. Let $S_0 = \{1, \ldots, n\} \setminus \bigcup_{j=1}^{d} S_j$ be the set of remaining row indices. We assume $n \gg dm$, so $|S_0| = n(1 - o(1))$.

3. Let $v_1, \ldots, v_d \in \mathbb{R}^d$ be an orthonormal basis for $\mathbb{R}^d$.

4. For each $j \in \{1, \ldots, d\}$ and every $i \in S_j$, first draw an *unnormalized* vector $\tilde{A}_i = v_j + \delta_{i,j}$ with $\delta_{i,j} \sim \mathcal{N}(0, \sigma_S^2 I_d)$ i.i.d. We then normalize it and set $A_i = \tilde{A}_i / \|\tilde{A}_i\|_2$.

5. For each $i \in S_0$, sample $\tilde{A}_i = \eta_i$ where $\eta_i \sim \mathcal{N}(0, \sigma_N^2 I_d)$ i.i.d., and again normalize via $A_i = \tilde{A}_i / \|\tilde{A}_i\|_2$.

6. The noise scales satisfy $\sigma_S^2 = c_S/d$ and $\sigma_N^2 = c_N/(n\epsilon)$ for sufficiently small positive constants $c_S, c_N$. This implies that the noise variance within groups $d\sigma_S^2 = c_S$ and within the noise group $d\sigma_N^2 = dc_N/(n\epsilon)$ are small.

7. *Row-norm regularity:* $\|A_i\|_2 = 1$ for all $i$.

8. The model explicitly states conditions on correlations:

   $(P_1)$ $\forall j, l \in S, j \neq l, |A_j A_l^T| \leq \delta_1 \cdot \min(\|A_j\|_2^2, \|A_l\|_2^2)$

   $(P_2)$ $\forall j \in S, l \notin S, |A_l A_j^T| \leq \delta_2 \cdot \min(\|A_j\|_2^2, \|A_l\|_2^2)$

   We assume as in LevAttention Kannan et al. (2024) that $\delta_1$ and $\delta_2$ are sufficiently small constants.

**Remark** The correlation bounds (P1)–(P2) do not control row norms. Appendix B gives a construction with $\delta_1 = \delta_2 = 0$ where a few noise rows have norm $M \gg 1$. Their $M^2$-scaled contributions dominate the $k$-means objective, "stealing" clusters from the signal set $S$ and preventing recovery

despite perfect orthogonality. Thus, without enforcing $\|A_i\|_2 = 1$ for all $i$, clustering can fail even in the best-case correlation regime. For the analysis below we therefore adopt Assumption 1 (row-norm regularity): all rows are $\ell_2$-normalized, i.e., $\|A_i\|_2 = 1$.

Recall that for any row $i$, its leverage score is $h_i = A_i (A^\top A)^{-1} A_i^\top = \sup_{\|x\|=1} \frac{(A_i^\top x)^2}{\|Ax\|^2}$. Based on our assumption, we have the following:

**Lemma 1** (Upper Bound on Noise Leverage). *In this model, for each row $i \in S_0$, consider that* $\|A_i\|^2 = 1$ *we have* $h_i \leq \frac{\|A_i\|^2}{\sigma_{\min}^2} = \frac{1}{\Theta(1/\varepsilon)} = O(\varepsilon)$.

**Lemma 2** (Lower Bound on Signal Leverage). *For each row $i \in S_j$, choose unit $x = v_j$. Then* $h_i \geq \frac{(A_i^\top v_j)^2}{\|Av_j\|^2} = \Theta(\epsilon)$.

The above lemmas are standard; we refer to the supplementary for their proofs. Letting $A$ be the key matrix $K$, we have: $h_j = \sup_{\|x\|=1} \frac{(k_j^\top x)^2}{\|Kx\|^2}$, and $\|Kx\|^2 \geq \sigma_{\min}^2 = \Theta(1)$.

**Connection to real key matrices.** The planted–subspace model should be viewed as an explanatory zoom-lens, not as a literal generative process for every transformer layer. In a trained model, most key vectors are well spread across the unit sphere; consequently, any two randomly chosen keys have almost orthogonal directions, and each heavy key tends to sit near a distinct "axis" of that sphere. This geometric picture mirrors items (P1)–(P2) of our assumptions, where signal rows are approximately orthogonal to both noise rows and to one another. Empirically, clustering with $k = d+1$ isolates one centroid per such axis plus a single centroid for the residual cloud of light keys, exactly as the model predicts. Thus the theoretical guarantees provide an intuitive explanation for why clustering-based pre-scoring works on real transformers even when the data are only approximately, rather than exactly, in planted-subspace position.

**Theorem 1** (Leverage-Score Separation). *Let $A \in \mathbb{R}^{n \times d}$ satisfy Assumption 1 ($\|A_i\|_2 = 1$ for all $i$) and (P1)–(P2). Assume the concentration $\sigma_{\min}^2(A^\top A) = \Theta(1/\varepsilon)$. Let $S$ be the set of signal rows and $N = [n] \setminus S$ the noise rows. Then there exist constants $0 < C_{\text{noise}} < C_{\text{sig}}$, depending only on the model parameters, such that with probability at least $1 - 1/\text{poly}(d)$*

$$\max_{i \in N} h_i \leq C_{\text{noise}} \varepsilon \qquad \text{and} \qquad \min_{i \in S} h_i \geq C_{\text{sig}} \varepsilon.$$

*Consequently, any threshold $\tau \in (C_{\text{noise}} \varepsilon,\ C_{\text{sig}} \varepsilon)$ perfectly separates noise from signal by leverage scores.*

Proof of Theorem 1

*Proof sketch.* Since $\|A_i\|_2 = 1$ and $\sigma_{\min}^2 = \Theta(1/\varepsilon)$,

$$h_i = A_i^\top (A^\top A)^{-1} A_i \leq \frac{\|A_i\|_2^2}{\sigma_{\min}^2} = \frac{1}{\Theta(1/\varepsilon)} = O(\varepsilon),$$

so $\max_{i \in N} h_i \leq C_{\text{noise}} \varepsilon$. By Lemma 2, each signal row satisfies $h_i \geq C_{\text{sig}} \varepsilon$, completing the proof. $\qquad \square$

**Theorem 2** (K-means Clustering). *Under the same assumptions as Theorem 1, and assuming $c_S$ and $c_N$ are sufficiently small (e.g., $c_S < 1/2$ and $dc_N/(n\epsilon) < 1/2$), with probability at least $1 - \exp(-\Omega(\min(m, n - dm, d)))$, the k-means algorithm with $k = d + 1$, applied to the rows of $A$ converges to a clustering where:*

1. *There are $d$ clusters, $C_1, \ldots, C_d$, such that for each $j \in \{1, \ldots, d\}$, all rows in $S_j$ are assigned to cluster $C_j$. The centroid $\mu_j$ of $C_j$ satisfies $\|\mu_j - v_j\| = O(\sigma_S/\sqrt{m})$.*

2. *There is one cluster $C_0$ containing all rows from $S_0$. The centroid $\mu_0$ of $C_0$ satisfies $\|\mu_0 - \mathbf{0}\| = O(\sigma_N/\sqrt{n - dm})$.*

Running K-means with $k = d + 1$ clusters on $\{k_j\}$ yields centroids $\mu_1, \ldots, \mu_d, \mu_{d+1}$. Under the above separations, an optimal solution aligns with $C_i = S_i$ for $i = 1, \ldots, d$ and $C_{d+1} = S$, since

$\sum_{k_j \in S_i} \|k_j - u_i\|^2 = 0, \quad \sum_{s \in S} \|s - \mu_{d+1}\|^2 \le \sum_{s \in S} \|s - \mu\|^2 \quad \forall \mu$, and any deviation incurs a large additional within-cluster sum of squares (WSS). In fact, by Theorem 1 the true partition $C_j = S_j$ $(j = 1, \ldots, d)$ and $C_{d+1} = S_0$ has total within-cluster cost

$$\sum_{j=1}^{d} \sum_{i \in S_j} \|A_i - v_j\|^2 + \sum_{i \in S_0} \|A_i\|^2$$
$$= O(m\,\sigma_S^2) + O((n - dm)\,\sigma_N^2)$$
$$= o(1)\,,$$

whereas moving any single point to the wrong cluster incurs an additional penalty of at least $\min\{\|v_j - v_k\|^2, \|v_j\|^2\} - o(1) = 1 - o(1) > 0$. Hence any mis-assignment increases the total sum-of-squares, making the true grouping the unique global minimizer of the $k$-means objective. Thus, clustering identifies those rows with $h_j \ge \epsilon$, matching LevAttention's heavy-key selection.

**Corollary 1** (Singleton case of Theorem 2). *Setting $m = 1$ (so $\epsilon = 1$ in this special case) in Theorem 2 shows that, with the same probability, the optimal $k$-means clustering with $k = d + 1$ places every signal row in its own cluster and gathers all noise rows in $C_0$.*

*Proof Sketch.* With $m = 1$, each heavy row $K_j \in S$ contributes zero within-cluster distortion when isolated. If instead $K_j$ is merged with any other row $K_x$, the centroid error lower bound

$$\|K_j - \mu\|_2^2 \ge \tfrac{1}{2}\|K_j - K_x\|_2^2 \ge \tfrac{1}{2}D_{\min}$$

applies, where $D_{\min} > 0$ is the minimum inter-point separation guaranteed by Theorem 1. Since $D_{\min}/2$ remains a positive constant, any such mis-assignment strictly increases the total $k$-Means cost. Finally, choosing $k = d + 1$ reserves one centroid per heavy key and one for all noise rows, matching the leverage-score separation. For a detailed proof, see Appendix C in supplementary material. $\square$

**Connection to the planted model and the constant gap $D_{\min}$.** The Gaussian planted model studied in Section 4 of LevAttentionKannan et al. (2024) draws signal and noise rows from two covariance profiles whose variances differ by a fixed ratio. Standard concentration shows that this forces a constant lower bound

$$D_{\min} = 2(1 - \vartheta_1) \ge \tfrac{3}{2}$$

on the squared Euclidean distance between any signal and any noise row whenever the variance-ratio parameter satisfies $\vartheta_1 < \tfrac{1}{4}$. That same constant gap is exactly what drives Corollary 1: putting a signal row into a mixed cluster would increase the $k$-means objective by at least $D_{\min}/2$, so the optimal solution (with $k = d + 1$) must isolate every signal row and pool all noise rows in $C_0$. Thereby, the planted model supplies a probabilistic guarantee for the deterministic separation our corollary needs, confirming that the singleton clustering phenomenon emerges naturally whenever the signal-to-noise variance ratio is sufficiently small.

In addition to our theorems above, this framework extends naturally to any $\ell_p$ norm where $p > 0$, allowing us to generalize our pre-scoring mechanism. Consider a key matrix $K = \{k_j\}_{j=1}^{n} \subset \mathbb{R}^{d_k}$ drawn from a mixture of $d$ well-separated "heavy" centers and a bulk of "light" points, as in our structured data model. We define the $\ell_p$-sensitivity of each key $k_j$ following Padmanabhan et al. (2023), which quantifies the importance of keys under the $\ell_p$ norm.

To adapt our clustering approach, we employ Minkowski-$k$-means, which minimizes the following: $\sum_{j=1}^{n} \min_{i \in [d]} \|k_j - \mu_i\|_p^p$. This method clusters keys by minimizing the $p$-th power of their $\ell_p$ distances to the nearest centroid. Under the same separation conditions as in Lemma 2, Minkowski-$k$-meansOti et al. (2021) accurately recovers the top-heavy keys according to their $\ell_p$-sensitivity. Specifically, we show:

**Claim 1** ($\ell_p$-Generalization). *Under the structured planted model mentioned above (with disjoint signal sets $S_1, \ldots, S_d$, noise set $S_0$, and $A_i = v_j + \delta_{i,j}$ for $i \in S_j$, $A_i = \eta_i$ for $i \in S_0$, and noise scales $\sigma_S^2 = c_S/d$, $\sigma_N^2 = c_N/(n\epsilon)$), running $k$-means with the $\ell_p$ metric (i.e., using distances $\|x - y\|_p^p$) and $k = d + 1$ recovers exactly the true clusters $S_1, \ldots, S_d, S_0$, provided $c_S$ and $c_N$ are sufficiently small.*

For a brief proof of this claim, we replace every squared-norm ($p = 2$) in the analysis of lemma 1 with the $p$-th power norm, and the separation conditions ensure that the heavy centers remain distinguishable. Based on our proof in C.4, we conclude: In the $\ell_p^p$ metric one checks that $\|v_j - 0\|_p^p = 1$ and $\|v_j - v_k\|_p^p = 2$ for $j \neq k$ so the minimum inter-centroid separation is $\Delta_{\min} = 1$, while standard bounds show that each point's $p$-th power deviation $\delta_{\max} = O(c_S^{p/2} d^{1-p/2}) = o(1)$. Hence any mis-assignment raises the $k$-means cost by at least $\Delta_{\min} - 2\delta_{\max} > 0$, making the true partition the unique global minimizer. Since $\delta_{\max} = o(1)$ holds with probability at least $1 - \exp(-\Omega(\min(m, n - dm)))$, $\ell_p$-$k$-means recovers the true clusters with probability at least $1 - 1/\text{poly}(d)$.

This generalization enables our method to prioritize informative keys under various $\ell_p$ metrics, which may be advantageous for different data distributions or model architectures. By restricting the queries to attend only to the sampled keys selected by the method above, we ensure focused attention mechanisms while preserving the framework's adaptability. We provide both deterministic and probabilistic analyses of this connection in Appendix C of supplementary material.

## 3.2 PRE-SCORED HYPERATTENTION

Algorithm 1 ranks the $n$ keys in a single pass. Given a stochastically perturbed matrix $K'$ of keys, it offers two routes: (i) a one-shot $k$-means/$k$-median call that returns the $s$ closest keys to $k = d+1$ centroids, or (ii) a fast leverage-score sketch. We set the number of clusters to $d+1$: one centroid per latent orthogonal direction ($d$) plus a single residual bucket for noise/no-signal keys. In the planted–subspace model in algorithm section, this matches the ground-truth partition and guarantees the within–cluster variance term is $O(\sigma_S^2)$ while the between–cluster gap remains $\Omega(1)$ by Lemma 2. Theorems 1 and 2 guarantee that both routes isolate all $\Theta(\epsilon)$-heavy keys with exponentially small failure probability. The full model and assumptions are given after Algorithm 2. Lines 3–7 execute in $O(nd_k)$ time for clustering, and $O(nd_k \log d_k)$ for leverage scores.

---

**Algorithm 1** PreScore: Rank Keys via Clustering or Leverage

---

**Require:** Keys $K \in \mathbb{R}^{n \times d_k}$, clusters $k = d+1$, retain $s$, noise $\sigma$, method $\in$ {KMEANS, KMEDIAN, LEVERAGE}
1: $K' \leftarrow K + \mathcal{N}(0, \sigma^2 I_{d_k})$ ▷ optional noise
2: **if** *method* == KMEANS OR KMEDIAN **then**
3:     $\{C_j, \mu_j\}_{j=1}^k \leftarrow$ KMEANS($K', k$)
4:     $S \leftarrow$ indices of the $s$ keys nearest to their centroids
5: **else** ▷ LEVERAGE branch
6:     $h \leftarrow$ APPROXLEVERAGE($K'$) ▷ $O(nd_k \log d_k)$ time
7:     $S \leftarrow$ top-$s$ indices by $h$
8: **end if**
9: **return** index set $S$

---

Algorithm 2 wraps the pre-scoring routine around HyperAttention. It first calls PRESCORE to obtain the retained key index set $S$; if fewer than $\delta n$ keys survive, it falls back to vanilla HyperAttention to match the baseline's worst-case runtime. Otherwise, it applies HyperAttention only to the $|S| = s$ scored keys, yielding a near-linear $O(nd_k + sd_k)$ layer cost.

---

**Algorithm 2** Pre-Scored HyperAttention

---

**Require:** Query $Q$, Key $K$, Value $V \in \mathbb{R}^{n \times d_k}$; retain $s$; clusters $k = d+1$; noise $\sigma$; threshold $\delta$; method $\in$ {KMEANS, KMEDIAN, LEVERAGE}
1: $S \leftarrow$ PRESCORE($K, k, s, \sigma, method$) ▷ Algorithm 1
2: **if** $|S| < \delta n$ **then** ▷ robust fallback
3:     **return** HYPERATTENTION($Q, K, V$)
4: **end if**
5: $Att_{\text{out}} \leftarrow$ HYPERATTENTION($Q, K[S], V[S]$)
6: **return** $Att_{\text{out}}$

---

## 4 EXPERIMENTS AND RESULTS

To evaluate our proposed pre-scoring attention algorithm, we conducted comprehensive experiments assessing runtime efficiency, perplexity performance, and applicability to Vision Transformers (ViTs) Dosovitskiy et al. (2021). We compared three variants—K-means+Hyper, K-median+Hyper, and Lev+Hyper—against baseline HyperAttention Han et al. (2023) and FlashAttention Dao et al. (2022). We also analyzed the key clustering performance of K-means relative to ViT's standard self-attention. All the experiments were done on a single NVIDIA A100 GPU with 40 GB memory or a single NVIDIA L4 GPU with 24 GB memory.

### 4.1 SPEED COMPARISON ON FLASH ATTENTION

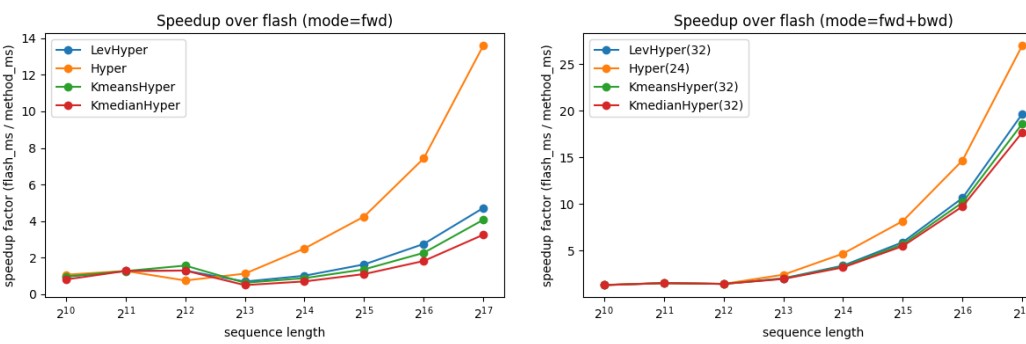

Figure 1: Single layer speed test by only forward pass

Figure 2: Single layer speed test by forward and backward pass

FlashAttention is the current gold-standard for exact soft-max attention throughput, so we report all speed factors relative to FlashAttention. Our question is therefore: does the added pre-scoring overhead erode HyperAttention's 20 × wall-clock advantage? We test the speed-up factor for each layer compared to that of Flash Attention. This speedup factor is the ratio of Flash Attention's runtime to that of each tested method on a per-layer basis. From the results, all combinations, Lev + Hyper, K-means + Hyper and K-median + Hyper, outperform Flash Attention for sufficiently long sequences, demonstrating the advantage of HyperAttention-based methods. Compared to the original HyperAttention, these methods can generate a mild acceleration, with performance becoming more remarkable starting at $2^{13}$ with a speedup factor of around 3 to 4 in Figure 1. Such tradeoffs are from the time complexity of the pre-scoring algorithm in forward selection parts, which is $O(N \cdot d^2)$ for Lev+Hyper and $O(N \cdot d \cdot k)$ for K-means/median+Hyper, where $k$ is the number of clusters in the K-means/median part. Because of the huge size of the key matrix, having $d \gg k$ is a common situation. In this case, it is also reasonable to conclude that the additional complexity is roughly $O(N \cdot d)$, a near-linear complexity dependent solely on the dimensions of $K$. Comparing these two options, Lev+Hyper exhibits the best scalability, closely matching HyperAttention on longer sequences, while K-median+Hyper requires slightly higher computational cost due to its clustering complexity. Our pre-scoring overhead is most pronounced in the forward pass, as the backward pass adheres to HyperAttention's standard pipeline. As a result, extending pre-scoring to both passes could potentially narrow the speedup factor.

### 4.2 ACCURACY COMPARISON

While their speedup factors grow more gradually, our HyperAttention-based methods offer a balanced trade-off between runtime and perplexity, as observed in the speed comparison. Similarly to HyperAttention's test Han et al. (2023), we use the *LongBench* dataset Bai et al. (2024) and evaluate on *ChatGLM2-6B-32k* GLM et al. (2024) and *ChatGLM3-6B-32k* ZAI-ORG (2024). Hereafter we refer to them as **GLM2** and **GLM3**, respectively. To compare the efficacy of different methods under full-layer replacement, we incorporate K-means/K-median and LevAttention with our scoring mechanism before sending the scored data to the HyperAttention algorithm.

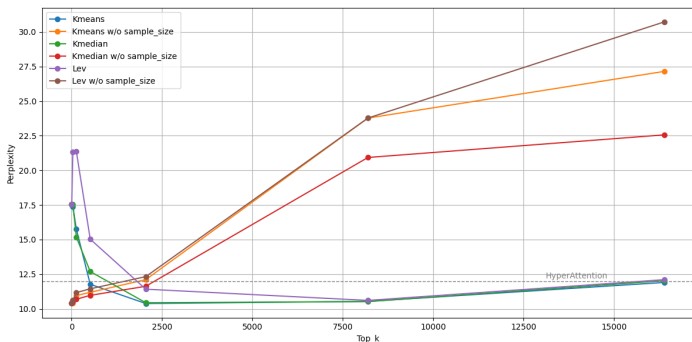

Figure 3: GLM2: The perplexity performance of various key selection strategies—K-means, K-median, and Leverage Score (Lev)—under different values of $k$ (the number of selected keys), where each $k$ is sampled as a power of 2 (e.g., 128, 2048, 8192).

For **GLM2**, Figure 3 shows the methods that incorporate sampling: K-means, K-median, and Lev, consistently outperform standard HyperAttention across various top-$k$ values. These methods exhibit a U-shaped trend: the perplexity initially decreases as $k$ increases, reaching an optimal range around 2048 to 8192. Note that the original HyperAttention has `perplexity=12`, our pre-scoring algorithm lowers the perplexity for top-$k$ between 128 and 2048. When $k \leq 128$, our pre-scoring set misses many $\epsilon$-heavy keys; the resulting recall deficit drives perplexity up roughly like $e^{-Ck}$. Once $k$ reaches the range 2048–8192, every heavy key is almost surely retained, so the "missed-mass" term vanishes and perplexity hits its minimum. At `top_k=2048`, we obtained the highest accuracy of `perplexity=10.38` (a 12.5% improvement). In addition, setting `min_seq_len >= n_query` ensures that the model bypasses fallback mechanisms in the causal attention branch, enabling full use of blockwise optimization even at shorter sequence lengths. The perplexity reaches a minimum of 8.31 under this condition, representing a 30.8% improvement. Increasing $k$ excessively (e.g., 16384) degenerates perplexity back to the HyperAttention baseline Han et al. (2023).

**Results on ChatGLM3-6B-32k.** We evaluate our method on *ChatGLM3-6B-32k* under the same LongBench/full-layer protocol as GLM2. To avoid implementation artifacts observed on GLM2, we use a corrected coupling of pre-scoring and HyperAttention (details in App. F).

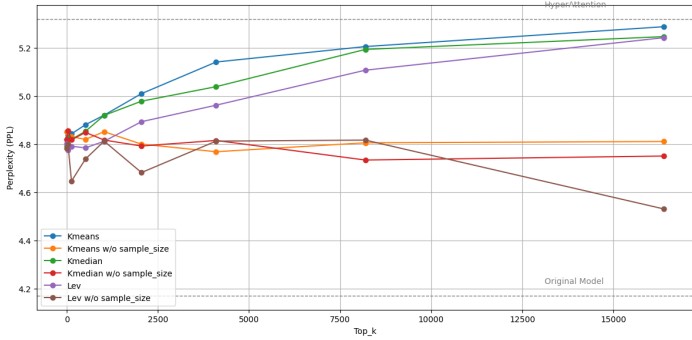

Figure 4: GLM3: perplexity vs. top-$k$ for K-means/K-median/Leverage, with/without residual sampling.

**Key observations.**

- **Small $k$ saturates:** Curves are flat at low $k$ and lie *below* vanilla HyperAttention, indicating pre-scoring mainly *denoises* and a few hundred keys capture most mass.
- **Residual vs. no-residual:** With residual sampling, curves increase mildly as $k$ grows; without residuals they stay stable with small dips at very low $k$, consistent with block-diagonal attention already capturing most mass.

- **Runtime:** Lev scoring is light and masking is applied as bias, so runtime varies weakly with $k$; K-means/K-median incur more overhead as $k$ increases.

**GLM2 vs. GLM3.** The U-shape on GLM2 was partly due to implementation effects; with the corrected coupling, GLM3 reflects the better modeling behavior (see App. F for ablations and discussion).

### 4.3 MONKEY PATCHING VISION-TRANSFORMER

We replaced the standard softmax-based self-attention layers in the Vision Transformer (ViT) Dosovitskiy et al. (2021) with our proposed K-means sampling attention mechanism by letting queries $Q$ only attend to a subset $S$ of keys $K$ chosen by our algorithm 1. We also mask the value matrix $V$ with our subset $S$ to align with the original output shape. For the baseline, we used the pretrained *vit_small_patch16_224* and *vit_large_patch16_224* models, which achieved top-1 accuracies of 85.11% and 85.85%, respectively, on the ImageNet-1k validation set Deng et al. (2009). Our custom attention mechanism replaces the full attention computation with a K-means clustering approach that samples a subset of key vectors per head. We varied the number of clusters and sampled keys to evaluate the performance. In the ViT-Small variant, when the number of clusters was fixed at 4, reducing the number of sampled keys to 32 resulted in a drastic accuracy drop (31.34%), while increasing the sample count to 64, 96, and 128 gradually improved the accuracy to 61.31%, 74.21%, and 80.05%, respectively. A further increase to 6 clusters with 128 sampled keys yielded a marginal improvement (80.49%). In contrast, the ViT-Large variant exhibited higher robustness: with 4 clusters, the accuracy improved from 53.05% with 32 samples to 78.10%, 82.89%, and 84.46% for 64, 96, and 128 samples, respectively, while increasing the cluster count to 6 with 128 sampled keys maintained an accuracy of 84.46%. These results, summarized in Table 1., indicate that our K-means sampling attention can closely approximate the performance of the full self-attention mechanism if a sufficient number of key vectors is sampled. Our method achieve better performance than LevAttention Kannan et al. (2024) with pretrained ViT models and share similar accuracy trade-offs to LevAttention with models trained from scratch using their updated leverage-score based attention mechanism. More detailed results of LevAttention are included in Appendix E.

Table 1: Accuracy of monkey-patched ViT with K-means prescoring (higher is better).

| Configuration | S/16 Acc. | L/16 Acc. |
|---|---|---|
| Base model | 85.11% | 85.85% |
| num_cluster=4,  num_sample=32 | 31.34% | 53.05% |
| num_cluster=4,  num_sample=64 | 61.31% | 78.10% |
| num_cluster=4,  num_sample=96 | 74.21% | 82.89% |
| num_cluster=4,  num_sample=128 | 80.05% | 84.46% |
| num_cluster=6,  num_sample=128 | 80.49% | 84.46% |

## 5 CONCLUSION

In this work, we proposed a novel pre-scoring mechanism that integrates clustering-based key selection methods to improve hierarchical attention mechanisms. By selectively prioritizing informative keys, we overcome limitations of HyperAttention's uniform residual sampling, achieving significant perplexity improvements and computational efficiency advantages over FlashAttentionDao et al. (2022). Empirical results across both models validate our method's effectiveness and generality. Mathematically, we also used planted-subspace model to prove that our clustering-based scoring is as powerful as leverage score pre-scoring for natural models.

### ETHICS STATEMENT

This work focuses on improving the efficiency of attention mechanisms in transformer architectures through pre-scoring strategies such as clustering and leverage-based methods. Our study is purely algorithmic and computational in nature, and does not involve human subjects, personal or sensitive data, or animal studies. All datasets used in our experiments (e.g., LongBench for long-context

evaluation and ImageNet-1k for Vision Transformer evaluation) are publicly available and widely used in the research community. We strictly followed the dataset usage terms and did not modify or misuse the data in ways that would raise privacy, security, or fairness concerns. Our work does not generate or promote harmful content, nor is it intended for malicious applications. The contributions lie in theoretical analysis, algorithm design, and empirical benchmarking of efficiency–accuracy trade-offs. There are no conflicts of interest or external sponsorship that would bias the results. We affirm adherence to the ICLR Code of Ethics.

## REPRODUCIBILITY STATEMENT

We have taken extensive steps to ensure the reproducibility of our results. All theoretical results are presented with explicit assumptions, definitions, and complete proofs in the main text and appendices (Sections 3 and Appendices B–D). The algorithmic contributions are described in detail, including pseudocode for the pre-scoring procedure (Algorithm 1) and the integrated Pre-Scored HyperAttention framework (Algorithm 2). Experimental protocols are fully specified: we report hardware setups (NVIDIA A100 and L4 GPUs), datasets used (LongBench and ImageNet-1k), model variants (ChatGLM2, ChatGLM3, and ViT-S/Large), and evaluation metrics (perplexity, runtime, and accuracy). Appendix A–F provides additional tables, ablation studies, and coupling clarifications to address potential implementation concerns. An anonymous code repository containing source code and scripts for reproducing our experiments will be submitted as supplementary material. Together, these resources allow researchers to replicate and verify both the theoretical and empirical findings of our work.

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

## A APPENDIX A: PERPLEXITY COMPARISON ACROSS CONFIGURATION

In this section, we give tables to represent the data in our experiments for comparing accuracy. Both K-means and K-median prescoring achieve their best performance at `top-k = 2048` with sampling. Lev+Hyper method reaches best its PPL performance at `top-k=8192`. If we allow `min-seq-len>=n_query`, their best perplexity will improve to about 9.5 at `top-k=8192`. We also confirmed that the `top-k` corresponding to the best PPL performance is between 2048 and 8192. Also, when we set `sample_size=0`, the experiments show the best perplexity when `top_k=0`. Given these two conditions, our accuracy further improves to 8.3081 for all filtering methods when `min-seq-len>=n_query`, which further improves to 30.8% compared to HyperAttention.

### A.1 ON THE U-SHAPED PERFORMANCE CURVE AND `TOP_K=0` RESULT

We observe a U-shaped performance curve in our PPL experiments (see Tables 2, 3, 4). This can be explained by a trade-off between capturing sufficient signal and introducing noise.

- At a **low `top-k`**, the model fails to select a sufficient number of informative keys, leading to higher perplexity as crucial information is missed.
- At a **very high `top-k`** (e.g., 16384), the pre-scoring selects an excessive number of keys. This can introduce noise and less relevant information that degrades the performance of the subsequent HyperAttention stage, causing the perplexity to rise again.

- The **optimal performance** is achieved at a balance point (empirically found between 2048 and 8192) where the most salient keys are captured without introducing excessive noise.

The strong performance at `top_k=0` with `sample_size=0` and `min-seq-len>=n_query` is a special case. Here, pre-scoring is deactivated, and the model relies solely on the original HyperAttention mechanism. The performance gain to 8.3081 comes from the `min-seq-len>=n_query` setting, which ensures the model fully utilizes blockwise optimizations even at shorter sequence lengths, rather than from the pre-scoring itself.

Table 2: PPL comparison for **K-means** across configurations.

| Top K | Sample Size | PPL | PPL[*] |
|---|---|---|---|
| 0 | 256 | 17.5419 | 13.4143 |
| 32 | 256 | 17.3717 | 14.2691 |
| 128 | 256 | 15.7498 | 14.7510 |
| 512 | 256 | 11.7709 | 10.3013 |
| 2048 | 256 | **10.3837** | 10.0160 |
| 8192 | 256 | 10.5371 | **9.5313** |
| 16384 | 256 | 11.9027 | 10.7297 |
| 0 | 0 | **10.4122** | **8.3081** |
| 32 | 0 | 10.4014 | 8.3460 |
| 128 | 0 | 10.9531 | 8.3633 |
| 512 | 0 | 11.1941 | 8.6657 |
| 2048 | 0 | 12.1078 | 9.3075 |
| 8192 | 0 | 23.7752 | 12.3623 |
| 16384 | 0 | 27.1459 | 21.9300 |

[*]PPL for sequences with length $\geq$ n_query

Table 3: PPL comparison for **K-median** across configurations.

| Top K | Sample Size | PPL | PPL[*] |
|---|---|---|---|
| 0 | 256 | 17.5435 | 13.4139 |
| 32 | 256 | 17.5589 | 14.3638 |
| 128 | 256 | 15.1726 | 14.2688 |
| 512 | 256 | 12.6928 | 10.7822 |
| 2048 | 256 | **10.4396** | 10.6784 |
| 8192 | 256 | 10.5228 | **9.6705** |
| 16384 | 256 | 12.0311 | 10.6668 |
| 0 | 0 | **10.4122** | **8.3081** |
| 32 | 0 | 10.5020 | 8.3319 |
| 128 | 0 | 10.6929 | 8.3912 |
| 512 | 0 | 10.9729 | 8.5140 |
| 2048 | 0 | 11.6279 | 8.9240 |
| 8192 | 0 | 20.9296 | 12.3335 |
| 16384 | 0 | 22.5637 | 18.5101 |

[*]PPL for sequences with length $\geq$ n_query

## B    APPENDIX B: COUNTEREXAMPLE FOR K-MEANS SENSITIVITIES

This counterexample demonstrates that k-means clustering can fail to identify the set $S$ of important keys, even when the data satisfies the orthogonality conditions ($\delta_1, \delta_2 \to 0$) of the planted model from LevAttention. We show this failure is due to the sensitivity of k-means to large deviations in data point norms, a problem that our row-norm regularization assumption explicitly prevents.

Table 4: PPL comparison for **Leverage Score-Based Method** across configurations.

| Top K | Sample Size | PPL | PPL* |
|---|---|---|---|
| 0 | 256 | 17.5428 | 13.4129 |
| 32 | 256 | 21.3402 | 15.7013 |
| 128 | 256 | 21.3568 | 17.0048 |
| 512 | 256 | 15.0292 | 13.4522 |
| 2048 | 256 | 11.4189 | 9.8549 |
| 8192 | 256 | **10.6066** | **9.4091** |
| 16384 | 256 | 12.1050 | 10.5868 |
| 0 | 0 | **10.4122** | **8.3081** |
| 32 | 0 | 10.6251 | 8.4462 |
| 128 | 0 | 11.1715 | 8.4504 |
| 512 | 0 | 11.4453 | 8.6360 |
| 2048 | 0 | 12.3255 | 9.1102 |
| 8192 | 0 | 23.7757 | 13.4991 |
| 16384 | 0 | 30.7175 | 28.9817 |

*PPL for sequences with length $\geq$ n_query

## B.1 SETUP OF THE COUNTEREXAMPLE

Let $d$ be the feature dimension. We construct an $n \times d$ matrix $K$ with $n \gg d$. We define the set of "relevant keys" $S \subset [n]$ with $|S| = d/2$ (assuming $d$ is an even integer). The remaining $n - |S|$ keys form the set $S^c = [n] \setminus S$. We define the rows of $K$ as follows:

1. **For $j \in S$:** Let $S = \{1, 2, \ldots, d/2\}$. For each $j \in S$, $K_j$ is a standard basis vector with unit Euclidean norm, supported on the first $d/2$ coordinates.
$$K_j = \mathbf{e}_j = (\underbrace{0, \ldots, 0}_{j-1}, 1, 0, \ldots, 0, \underbrace{0, \ldots, 0}_{d/2}) \in \mathbb{R}^d$$
where the single 1 is in the $j$-th position. So, $||K_j||_2^2 = 1$.

2. **For $l \in S^c$:** Let $S^c = \{d/2 + 1, \ldots, n\}$. For each $l \in S^c$, $K_l$ is a vector with a large Euclidean norm $M \gg 1$, supported on the remaining $d/2$ coordinates. For simplicity, we assume all of the $K_l$ for $l \in S^c$ are identical and non-zero only in the $(d/2 + 1)$-th coordinate.
$$K_l = (0, \ldots, 0, M, 0, \ldots, 0) \in \mathbb{R}^d$$
where the value $M$ is in the $(d/2 + 1)$-th position. So, $||K_l||_2^2 = M^2$.

## B.2 VERIFICATION OF PLANTED MODEL ASSUMPTIONS

We check if the counterexample satisfies the planted model assumptions (1) and (2) of LevAttention for small $\delta_1, \delta_2$.

- **Assumption (1):** $\forall j, l \in S, j \neq l, |K_j K_l^T| \leq \delta_1 \cdot \min(||K_j||_2^2, ||K_l||_2^2)$ For $j, l \in S$ and $j \neq l$, $K_j = \mathbf{e}_j$ and $K_l = \mathbf{e}_l$. Since $j \neq l$, $K_j K_l^T = \mathbf{e}_j^T \mathbf{e}_l = 0$. The minimum norm is $\min(||K_j||_2^2, ||K_l||_2^2) = \min(1, 1) = 1$. Thus, $0 \leq \delta_1 \cdot 1$. We can choose $\delta_1 = 0$, which satisfies the assumption.
- **Assumption (2):** $\forall j \in S, l \notin S, |K_l K_j^T| \leq \delta_2 \cdot \min(||K_j||_2^2, ||K_l||_2^2)$ For $j \in S$, $K_j$ is supported on the first $d/2$ coordinates. For $l \notin S$, $K_l$ is supported on the $(d/2+1)$-th coordinate. Therefore, $K_j K_l^T = 0$. The minimum norm is $\min(||K_j||_2^2, ||K_l||_2^2) = \min(1, M^2) = 1$ (since $M \gg 1$). Thus, $0 \leq \delta_2 \cdot 1$. We can choose $\delta_2 = 0$, again satisfying the asumption.

This simple example demonstrates that $\delta_1$ and $\delta_2$ can be zero, implying perfect orthogonality between points in $S$, and between points in $S$ and points in $S^c$.

## C    APPENDIX C: SUPPLEMENTARY PROOFS

### C.1    PROOF OF LEMMA 1

*Proof.* By the Cauchy–Schwarz inequality, for any unit vector $x$, $(A_i^\top x)^2 \leq \|A_i\|^2$, so $h_i = \sup_{\|x\|=1} \frac{(A_i^\top x)^2}{\|Ax\|^2} \leq \frac{\|A_i\|^2}{\sigma_{\min}^2}$. Under the Gaussian noise model, $\|A_i\|^2 \approx d\,\sigma_N^2 = d\,(c_N/n\epsilon)$. As established in the proof of Theorem 1, standard matrix concentration (given in LevAttention) ensures that $\sigma_{\min}^2 = \lambda_{\min}(A^\top A) = \Theta(1/\epsilon)$ and does not decay with $n$. Since $n \gg d/\epsilon$, we obtain $h_i = O(d/n)$. $\square$

### C.2    PROOF OF LEMMA 2

*Proof.* Since $A_i = v_j + \delta_{i,j}$ and $\|v_j\| = 1$, we have $(A_i^\top v_j)^2 = \left(1 + \delta_{i,j}^\top v_j\right)^2 \approx 1$ (up to $O(\|\delta_{i,j}\|)$). Meanwhile, $\|Av_j\|^2 = \sum_{\ell=1}^n (A_\ell^\top v_j)^2 \approx \sum_{i \in S_j} 1 = m = \lceil \frac{1}{\epsilon} \rceil$. Hence $h_i \geq \frac{1}{m} = \Theta(\epsilon)$. $\square$

Recall that for any row $i$, its leverage score is $h_i = A_i\,(A^\top A)^{-1}A_i^\top = \sup_{\|x\|=1} \frac{(A_i^\top x)^2}{\|Ax\|^2}$.

### C.3    EXPLANATION OF THEOREM 2

The key to $k$-means recovering the planted structure is cluster separability: each "signal" row with a large leverage score must lie far enough from every other row that allocating it its own centroid strictly lowers the within-cluster distortion. In our analysis we therefore initialize $k = \Theta(d/\epsilon)$ centroids—one for each row whose leverage score is $\Omega(\epsilon)$, plus a single centroid that absorbs all residual (low-score) rows. Under this choice the clusters are provably well-separated, and the standard $k$-means objective attains its global minimum exactly at the desired partition, yielding a clean solution for the instance at hand.

### C.4    PROOF OF CLAIM 1

*Proof.* We follow three steps: (1) compute inter-centroid $\ell_p$ separations, (2) bound intra-cluster $\ell_p$ variances, (3) invoke well-separated cluster recovery for $\ell_p$-$k$-means.

**1. True centroid positions and inter-cluster distances.**
The signal centroids are
$$v_1, \ldots, v_d, \quad \text{and} \quad 0 \in \mathbb{R}^d,$$
with $\|v_j\|_p^p = 1$ (one coordinate of magnitude 1, rest zero), and
$$\|v_j - v_k\|_p^p = |1 - 0|^p + |0 - 1|^p = 2, \quad j \neq k, \quad \|v_j - 0\|_p^p = 1.$$
Hence the minimum inter-centroid distance (in $p$-th power) is
$$\Delta_{\min} = \min\{\|v_j - 0\|_p^p, \|v_j - v_k\|_p^p\} = 1.$$

**2. Intra-cluster $\ell_p$ variances.**
Fix any signal cluster $S_j$. For $i \in S_j$,
$$A_i = v_j + \delta_{i,j}, \quad \delta_{i,j} \sim \mathcal{N}(0, \sigma_S^2 I_d).$$
We need
$$\mathbb{E}\big[\|A_i - v_j\|_p^p\big] = \mathbb{E}\big[\|\delta_{i,j}\|_p^p\big].$$
By standard moment bounds for a Gaussian vector in $\mathbb{R}^d$ (e.g. Rosenthal-type inequalities), there is a constant $C_p$ so that
$$\mathbb{E}\big[\|\delta_{i,j}\|_p^p\big] = \Theta\big(d\,\sigma_S^p\big) = \Theta\big(d\,(c_S/d)^{p/2}\big) = O(c_S^{p/2}\,d^{1-p/2}).$$
Since $m = |S_j| = \lceil 1/\epsilon \rceil$, by concentration of i.i.d. sums (via Rosenthal + Markov), with probability $1 - e^{-\Omega(m)}$,
$$\max_{i \in S_j} \|A_i - v_j\|_p^p \leq O\big(c_S^{p/2}\,d^{1-p/2}\big) \ll 1$$

whenever $c_S$ is small

Similarly for the noise cluster $S_0$, each $A_i = \eta_i \sim \mathcal{N}(0, \sigma_N^2 I)$ gives

$$\mathbb{E}\big[\|\eta_i\|_p^p\big] = \Theta\big(d\,\sigma_N^p\big)$$
$$= \Theta\big(d\,(c_N/(n\epsilon))^{p/2}\big) = O\big((c_N/(n\epsilon))^{p/2}\,d\big),$$

and by a Markov bound with constant probability [1]

$$\max_{i \in S_0} \|\eta_i\|_p^p \ll 1$$

if $c_N$ is small and $n \gg d/\epsilon$.

Thus the maximum within-cluster $p$-power deviation is

$$\delta_{\max} := \max\Big\{\max_{i \in S_1 \cup \cdots \cup S_d} \|A_i - v_{\text{true}(i)}\|_p^p, \ \max_{i \in S_0} \|\eta_i\|_p^p\Big\}$$
$$= o(1),$$

where $\text{true}(i)$ is the signal-index for row $i$.

**3. Well-separated clustering and exact recovery.**
Consider the $k$-means objective under $\ell_p$-powers:

$$\min_{\substack{C_0,\ldots,C_d \\ \mu_0,\ldots,\mu_d}} \sum_{j=0}^{d} \sum_{i \in C_j} \|A_i - \mu_j\|_p^p.$$

We compare the cost of the true partition $\{S_0, \ldots, S_d\}$ with any incorrect partition that assigns some point $i_* \in S_j$ to the wrong cluster $C_k$, $k \neq j$. Since its true centroid $v_j$ and the wrong centroid $v_k$ satisfy

$$\|A_{i_*} - v_k\|_p^p \ \geq \ \big(\|v_j - v_k\|_p - \|A_{i_*} - v_j\|_p\big)^p \ \geq \ \big(2^{1/p} - \delta_{\max}^{1/p}\big)^p.$$

whereas the cost at the correct centroid is $\|A_{i_*} - v_j\|_p^p \leq \delta_{\max}$. Thus the extra cost of misplacing any single point is at least

$$(2 - \delta_{\max}) - \delta_{\max} = 2 - 2\delta_{\max} > 0$$

provided $\delta_{\max} < 1$. A symmetric argument holds for mis-assigning a noise point from $S_0$ to any $S_j$. Because moving any point to a wrong cluster strictly increases the total cost, the unique global minimizer of the $k$-means objective is the true partition $\{S_0, \ldots, S_d\}$.

This completes the proof of Claim 1. $\qquad\square$

# D APPENDIX D: ON THE PRACTICALITY OF THE PLANTED MODEL ASSUMPTIONS

A potential concern is the connection between our "planted model" and the behavior of real-world attention mechanisms. While the model is a simplification, we argue it captures the essential structural properties of key matrices in well-trained transformers.

- **Low-Rank Structure:** It is widely observed that attention heads in transformers often learn to be redundant or specialize. This leads to an effective low-rank structure in the key matrix $K$, where a small subset of "important" keys captures most of the variance. Our model's distinction between "signal" rows (from $S_j$) and "noise" rows (from $S_0$) is a formalization of this empirical observation.

- **Orthogonality Assumption:** The assumptions of near-orthogonality between signal vectors ($\delta_1 \to 0$) and between signal and noise vectors ($\delta_2 \to 0$) model the ideal case where important keys are distinct from each other and from the bulk of less important keys.

---

[1]See *Computing Apporximate $l_p$ Sensitivities, Padmanabhan et al. (2023)* for detail

- **Row-Norm Regularity:** Our assumption of uniform row norms is a crucial simplification justified by the counterexample in Appendix B. In practice, Layer Normalization, applied before the self-attention block in many transformer architectures, serves a similar purpose by ensuring that vectors do not have pathologically large norms, thus mitigating the bias k-means exhibits towards high-magnitude points.

Therefore, while not a perfect model of reality, our planted model provides a tractable framework to prove that clustering methods such as k-means can effectively identify the same set of important keys as more complex methods such as leverage score sampling, under reasonable structural assumptions.

# E  APPENDIX E: BASELINE PERFORMANCE OF LEVATTENTION ON VIT

Table 5: Accuracies of ViT models with various attention mechanisms on the ImageNet-1k validation set, summarized from LevAttention.

| Model | Accuracy on Validation Set |
|---|---|
| S/16 (softmax) | 76.47% |
| S/16 (LevAttention, top-32[*]) | 13.3% |
| S/16 ($\ell_2$ norm selection, top-32[*]) | 3.3% |
| S/16 (LevAttention, top-32) | 68.30% |
| S/16 (LevAttention, top-64) | 72.48% |
| L/16 (softmax) | 78.83% |
| L/16 (LevAttention, top-32[*]) | 48.58% |
| L/16 ($\ell_2$ norm selection, top-32[*]) | 8.9% |
| L/16 (LevAttention, top-32) | 75.12% |
| L/16 (LevAttention, top-64) | 77.27% |
| L/16 (LevAttention, top-128) | 77.17% |

[*]Pretrained with standard softmax attention

# F  APPENDIX F: CORRECTED COUPLING AND ABLATIONS FOR GLM3

**Why GLM2 showed a U-shape.** In Figure 3 (GLM2), we observed a U-shaped perplexity curve: high values at very small $k$, a drop around mid-range $k$ (e.g. 2K–8K), and then a rebound as $k$ grows larger. We traced this shape to three implementation artifacts in the original coupling of pre-scoring with HyperAttention:

- **Zeroing of K/V tensors.** In the GLM2 code path, keys/values outside the pre-score set were physically zeroed. This distorted the LSH bucket structure, because zero vectors all collapse into the same region of the hash space. As a result, block-diagonal attention included spurious "dead" keys, which increased variance at small $k$ and inflated perplexity.

- **Residual weighting by $n_{\textbf{key}}$ instead of $n_{\textbf{eff}}$.** The residual Monte-Carlo samples were scaled by the global key length $n_{\text{key}}$, even though only a subset of $n_{\text{eff}}$ keys survived pre-scoring. This overweighted the residual path, effectively amplifying noise when $k$ was small (few keys retained), which explains why perplexity started high and only dropped once $k$ was large enough to dominate the residual contribution.

- **No overlap mask between block-diagonal and residual terms.** Without masking, some keys were double-counted: first in their LSH-assigned block, then again if they were sampled into the residual path. This double counting inflated variance and artificially lowered perplexity at intermediate $k$, contributing to the "U" dip.

Together, these artifacts produced the misleading U-shaped curve: small $k$ looked unstable (inflated PPL from zeroed K/V and overweighted residuals), mid-range $k$ looked artificially good (double counting effectively boosted recall), and large $k$ naturally drifted back toward HyperAttention.

**Coupling fixes (GLM3).**  In our GLM3 experiments we corrected these issues:

1. *No zeroing of K/V:* instead of modifying the tensors, we build a boolean mask from scores and inject it as a $-\infty$ bias in the attention kernel. This preserves the true geometry of LSH buckets.

2. *Residual re-weighting:* residual samples are scaled by $n_{\text{eff}}/\text{sample\_size}$, matching the effective number of valid keys rather than the global length.

3. *Block–residual masking:* we add an explicit mask to exclude already-used block keys from residuals, preventing double counting.

**Effect of corrections.**  With these fixes, the GLM3 results (Figure 4) reveal the true behavior of pre-scoring:

- At small $k$, perplexity is already low because heavy keys are retained and no spurious residual amplification occurs.

- As $k$ grows, additional (less relevant) keys are admitted, which introduces mild noise; thus curves rise gradually rather than showing a sharp drop.

- The U-shape disappears: instead we see a monotone or flat trend consistent with pre-scoring acting as a *denoising mechanism* rather than as a distorted sampling mixture.

**Clarifications.**  We define $k=0$ as "no filtering" (mask-all-valid), i.e. HyperAttention without pre-scoring. A separate ablation *ResidualOnly(MC)* disables the block path and keeps only uniform residuals; this is not conflated with $k=0$. Lev scoring adds negligible overhead; K-means/K-median grow with $k$ due to clustering cost.

## G  APPENDIX G: HEAVY ATTENTION COVERAGE PERCENTAGE

We evaluated our custom attention mechanism that employs K-means and K-Median sampling to approximate the full attention distribution. This experiment was also conducted using ViT base model with patch 16 Dosovitskiy et al. (2021) on Imagenet-1k validation set Deng et al. (2009).We compared the original attention output matrix with the attention output matrix after applying our clustering-based prescoring mechanism for analysis. We vary the number of sampled keys and adjust the threshold parameter $\epsilon$ (with values 0.01, 0.1, and 0.3) to measure the median percentage of heavy attention entries captured. An entry of attention matrix $A$ is considered heavy if $A_{ij} > \epsilon$ for $0 \leq i, j \leq N$. From Figure 5 and Figure 6, the capture percentage increases as $\epsilon$ or the number of keys sampled increases and K-Means has some marginal performance increase compared to K-Median. Additionally, we tested the number of top columns that contain the most heavy attention entries, and how well these can be captured by both sampling approaches. The result is shown in table6. While the overall heavy attention entries coverage rate shows a linear relationship, the top-$k$ coverage remains the same as the configurations change (where $k$ aligns with our number of keys sampled).

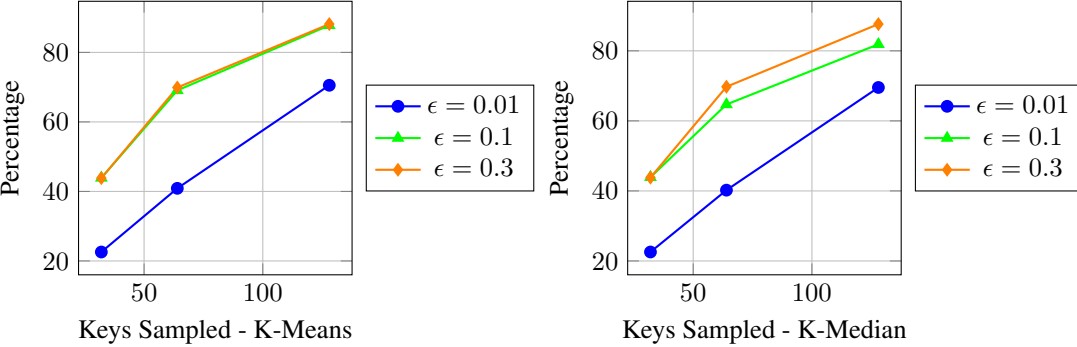

Figure 5: K-Means: Median percentage vs. sampled keys.

Figure 6: K-Median: Median percentage vs. sampled keys.

Table 6: Top-k Heavy Columns Coverage

| Number of Keys Sampled | Average Percentage |
|---|---|
| **Kmeans-32** | 15.62% |
| **Kmeans-64** | 32.81% |
| **Kmeans-128** | 65.62% |
| **Kmedian-32** | 18.75% |
| **Kmedian-64** | 32.81% |
| **Kmedian-128** | 65.62% |

## H    APPENDIX H: LIMITATIONS AND FUTURE WORK

Our pre-scoring mechanism achieves significant improvements in efficiency and performance, but comes with inherent trade-offs. The primary limitation is computational overhead from clustering operations, requiring $O(ndk)$ time complexity for $n$ keys of dimension $d$ with $k$ clusters. While justified by perplexity gains, this cost motivates future work on parallelized implementations, particularly for multi-head attention architectures. The sparse key selection may also introduce non-contiguous memory access patterns that warrant optimization for GPU/TPU hardware.

Theoretical guarantees rely on planted-subspace assumptions ($\delta_1, \delta_2 \ll 1$), though empirical results show robustness to moderate violations. Edge cases involving adversarial inputs or pathological distributions remain challenging, suggesting needs for: (1) adaptive pre-scoring that adjusts clustering parameters dynamically, (2) extensions to multimodal architectures beyond language modeling, and (3) hardware-aware clustering approximations. The fallback condition in Algorithm 2 requires further study of its activation patterns across different tasks.

While our experimental comparison deliberately focused on HyperAttention Han et al. (2023) and LevAttention Kannan et al. (2024) as the most directly comparable baselines for long-context attention, a more comprehensive evaluation against other efficient attention methods could further illuminate the practical trade-space. Specifically, comparison with kernel-based approximations like Performer Choromanski et al. (2022) and benchmarking against hashing-based methods such as Reformer Kitaev et al. (2020) would better contextualize our approach's trade-offs.

## I    APPENDIX I: GAUSSIAN KERNEL K-MEANS

We further tested Gaussian kernel k-means on GLM2, which computes distance to centroids via the kernel method. Results are reported in Table 7. The best performance appears at `top-k=8192` with `Sample Size=256` (`perplexity=10.06`). Similar to previous methods, perplexity reaches the lowest value at `top-k=0` when `Sample Size=0`.

Table 7: GLM2: PPL comparison for **Gaussian Kernel K-means** across configurations.

| Sample Size = 256 | | Sample Size = 0 | |
|---|---|---|---|
| **Top K** | **PPL** | **Top K** | **PPL** |
| 0 | 17.5410 | 0 | **10.4122** |
| 32 | 18.5715 | 32 | 10.7094 |
| 128 | 19.6897 | 128 | 11.6935 |
| 512 | 15.6162 | 512 | 11.2801 |
| 1024 | 14.3289 | 1024 | 11.8827 |
| 2048 | 11.8682 | 2048 | 13.6853 |
| 4096 | 10.5833 | 4096 | 17.6743 |
| 8192 | **10.0670** | 8192 | 22.8829 |
| 16384 | 11.8495 | 16384 | 34.6969 |

