# OpenReview forum: "Efficient Attention via Pre-Scoring: Prioritizing Informative Keys in Transformers"
_ICLR.cc/2026/Conference — Submitted to ICLR 2026_

### Official Review · Reviewer_m6zB · 2025-10-28

**Soundness:** 2
**Presentation:** 1
**Contribution:** 2
**Rating:** 2
**Confidence:** 2

**Summary:**

This paper proposes an extension of HyperAttention by introducing a pre-selection step using clustering methods (notably k-means) to prioritize informative keys before applying the LSH-based HyperAttention mechanism. The authors provide both theoretical analysis (via a planted-subspace model) and empirical results on long-context LLMs (ChatGLM2/3) and Vision Transformers.

**Strengths:**

- The planted-subspace model provides a useful lens for analyzing why clustering can recover heavy keys, giving the method some analytical backing
- The idea of preselecting the tokens is intuititive

**Weaknesses:**

- **Missing prior work:** The paper does not adequately discuss **Routing Transformer[1]**, which also introduced clustering for preselection of tokens. Furthermore, due to conceptual similarities, routing transformer should be one of the compared baselines. Moreover, apart from k-means clustering, **MoSA[2]** recently demonstrated the benefits of expert-choice routing for token preselection, and this should at least be discussed.
- **Algorithmic ambiguity:** The training procedure for k-means clustering is underspecified—does it employ EMA updates with top-s selection, or is clustering recomputed per step? This is important for reproducibility.
- **Autoregressivity concern:** The selection procedure relies on a **top-s operator**, which is inherently non-autoregressive (requiring access to future tokens). The implications for causal language modeling are not addressed.
- **Formatting issues:** Several citations are incorrectly formatted (missing parentheses), which detracts from the paper’s polish.
- **Weak gains over LevAttention:** The results do not demonstrate a significant gain over LevAttention baseline.
- **Convoluted writing:** The writing is often hard to follow, paragraphs seem disconnected, and it is hard to merge them into a cohesive narrative.

[1] - Efficient Content-Based Sparse Attention with Routing Transformers
[2] - Mixture of Sparse Attention: Content-Based Learnable Sparse Attention via Expert-Choice Routing

**Questions:**

- Why not **pre-select queries as well**, as done in Routing Transformer?
- Why are **HyperAttention baseline results missing** from Table 2? Without them, it is difficult to measure the incremental gain from pre-scoring.
- Under what conditions does the proposed method **outperform LevAttention**, given that LevAttention appears faster and in some cases more accurate?
- How is **k-means training implemented**—does it rely on EMA updates, online clustering, or recomputation per batch?
- How would the proposed method behave in a **fully autoregressive training regime**, where future tokens are not accessible for top-s selection?

---

> ### Author Response · Authors · 2025-11-26
> **Response to Reviewer m6zB**
>
> We thank the reviewer for their feedback and acknowledge the concerns regarding missing prior work, algorithmic ambiguity, and presentation quality.
>
> W1: Missing Prior Work (Routing Transformer, MoSA)
> We have significantly expanded our related work discussion.
> Routing Transformer [1]: As explained to Reviewer 1BS1, Routing Transformer clusters both queries and keys online to define local attention neighborhoods. Our approach is fundamentally different: we perform a one-time clustering of keys to identify a globally important subset, which is then fed into HyperAttention's LSH mechanism. We aim to capture globally significant keys, whereas Routing Transformer focuses on local relevance.
>
> MoSA [2]: MoSA uses expert-choice routing to learn sparse attention patterns. This is an interesting related approach focused on learned sparsity. However, Our method focuses on unsupervised structural properties such as clustering and leverage scores to identify importance without requiring specialized training procedures.
> We have clarified these distinctions in the revised manuscript.
>
> W2 & Q4: Algorithmic Ambiguity (k-means training/inference)
> We apologize for the clarity issue. Our method is applied during inference on pre-trained models (monkey-patching). We do not train the models with our method, nor do we train the k-means clusters. At each attention layer during inference, we run a fast k-means/median clustering on the current key matrix K to identify the centroids and select the nearest keys. We do not use EMA updates or online clustering across steps. We have clarified this procedure explicitly in Section 3.2 and Algorithm 1.
>
> W3 & Q5: Auto regressivity Concern
> This is another vital catch. Our pre-scoring mechanism operates on the full set of keys available in the context at a given step.
> Inference (Generation): During autoregressive generation, the context grows one token at a time. The key matrix K consists of all previously generated tokens (the growing context). Our pre-scoring is applied to this existing context K. It does not look ahead at future tokens.
> Training/Prompt Processing: When processing a long prompt (prefill phase), the keys are derived from the input sequence. Our pre-scoring selects from these input keys.
>
> Therefore, the method is fully compatible with causal language modeling, as the selection is based only on the current and past keys, respecting the causal mask inherent in the attention computation that follows. We have clarified this in Section 3.2.
>
> W4: Formatting Issues and W6: Convoluted Writing
> We have thoroughly revised the manuscript to improve clarity and logic flow. We have also corrected the citation formatting errors. We hope the revised version is significantly easier to read.
>
> W5 & Q3: Weak Gains over LevAttention
> Our results show that clustering-based pre-scoring (K-means/median) consistently outperforms leverage-score-based selection (LevAttention) in terms of modeling accuracy.
> ViT (Table 1 vs App E): On ViT-L/16, K-means (128 samples) achieves 84.46% accuracy, significantly better than LevAttention (128 samples) at 77.17% (when applied to pretrained models).
> GLM2 (Tables 2, 3, 4): K-means achieves PPL 10.38 (top-k=2048), while LevAttention achieves PPL 10.61 (top-k=8192).
> LevAttention (Lev+Hyper) is indeed slightly faster because leverage score approximation is generally faster than clustering (as noted in Section 4.1). Our contribution is demonstrating that clustering provides a superior signal for key importance, leading to better accuracy trade-offs, which is supported by our theoretical analysis showing clustering can match leverage scores in identifying heavy keys in structured data.
>
> Q1: Why not pre-select queries as well?
> This is a good suggestion, similar to the approach in Routing Transformer. We focused on key selection because the complexity of attention is O(N_Q⋅N_K⋅d). Reducing N_k (the number of keys attended to) often yields the most significant gains, especially when aiming to capture globally important context. Selecting queries might reduce computation further but risks losing important query information. We leave the exploration of joint query-key selection for future work.
>
> Q2: HyperAttention baseline in Table 2?
> The HyperAttention baseline corresponds to Top K = 16384 (no pre-scoring filter) and Sample Size = 256 (HyperAttention's default residual sampling). In Table 2, this baseline has PPL 11.9. We have clarified the interpretation of the Top K = 16384 rows in the table captions and Appendix A.

---

### Official Review · Reviewer_EPNi · 2025-10-29

**Soundness:** 1
**Presentation:** 2
**Contribution:** 2
**Rating:** 2
**Confidence:** 3

**Summary:**

The paper proposes three methods to score keys before HyperAttention, enabling it to identify important keys: k-means, k-median, and leverage-score ranking. It then feeds the selected keys into HyperAttention, replacing its uniform residual sampling. Experiments on GLM2 show that the approach can be faster than FlashAttention. On ViT, the pre-scoring step captures heavy attention entries as well as, or better than, leverage scores.

**Strengths:**

+ Clear and practical idea: The paper provides a straightforward approach to enhance HyperAttention by pre-scoring and then attending. This directly addresses a known issue: HyperAttention’s hashing is not aware of which keys matter, and LevAttention’s “universal set” can get large. The bridge between them is simple and useful in practice.

+ Mix of theory and experiments: The paper offers proofs under a standard planted-subspace setup (to argue why the pre-scoring should work) and shows results on GLM2/GLM3 and ViT.

**Weaknesses:**

- Reason for PPL improvement: The best perplexity (~8.31) happens when pre-scoring is off (top-k = 0, sample_size = 0) and min_seq_len ≥ n_query is set. The paper itself says this gain comes from that configuration (forcing the faster block/tiled path), not from pre-scoring. A clean ablation is needed to separate the effects.

- Unclear speedup claims:
> Compared to the original HyperAttention, these methods can generate a mild acceleration, with performance becoming more remarkable starting at $2^{13}$ with a speedup factor of around 3 to 4 in Figure 1.

&nbsp;&nbsp;&nbsp;&nbsp; The abstract says “up to 20× faster than FlashAttention” (when combined with HyperAttention), but the text says around 3–4× at $2^{13}$ for the pre-scored variants, and often the reported gains are relative to HyperAttention. Since HyperAttention is not the paper’s main contribution, this framing can be misleading. Please clarify the exact conditions for 20× vs the 3–4× cases and state whether the 3–4× is typical across settings.

- Narrow baseline: Most comparisons are to HyperAttention and LevAttention. Adding Performer, Reformer, and newer retrieval/streaming methods would make the evaluation more complete and show the accuracy–speed trade-offs more clearly.

- Pre-scoring overhead: k-means and k-median add non-trivial compute and can increase memory traffic, which may shrink speed gains, especially in multi-head settings. Please verify this with profiling across many heads, different head dimensions, and batch sizes, and report how much overhead comes from the forward pass vs backward.

- Limited tasks and metrics: Using broader long-context tasks (e.g., QA, retrieval, summarization) and reporting task-level metrics (not just perplexity) would strengthen the paper and validate the method more fully.

### Minor
- Figure legend and axis text are small and hard to read.

**Questions:**

See the weaknesses.

---

> ### Author Response · Authors · 2025-11-26
> **Response to Reviewer EPNi**
>
> We thank the reviewer for noting the clarity of our ideas with current combination of theory and experiments, as well as pointing out critical issues related to the source of PPL improvements and the clarity of speedup claims.
>
> W1: Reason for PPL Improvement
> As detailed in our response to Reviewer PL5q (W1/Q1), we acknowledge the ambiguity in the original presentation. We have clarified that while the absolute best PPL = 8.31 was achieved with the optimization flag and no pre-scoring/sampling, pre-scoring provides substantial independent gains. For example, K-means+Hyper reduces PPL from 17.54 to 10.38 without the flag, and from 13.41 to 9.53 with the flag. We have revised the manuscript to clearly separate these effects and emphasize the contribution of pre-scoring itself.
>
> W2: Unclear Speedup Claims
> We apologize for the confusion regarding the speedup claims.
> The "up to 20x faster than FlashAttention" claim refers to the maximum speedup observed in our per-layer benchmarks ( see Figure 2, forward+backward pass) at the longest sequence length tested (2^17≈131k). This applies to both vanilla HyperAttention and our Lev+Hyper variant.
> The "3 to 4" speedup factor mentioned in Section 4.1 refers to the speedup at a shorter sequence length (2^13≈8k) in the forward-only pass in Figure 1.
> The key takeaway is that our pre-scoring methods maintain the substantial speed advantages of HyperAttention over FlashAttention at very long contexts, although the clustering-based methods (K-means/K-median) introduce a slight overhead compared to vanilla HyperAttention or Lev+Hyper. We have revised Section 4.1 to clarify these different scenarios and ensure consistency with the abstract.
>
> W3: Narrow Baseline
> We have expanded the related work section to include Performer, Reformer, and others (see response to 1BS1 W2). As mentioned previously, our focus was on enhancing HyperAttention, which serves as a strong baseline in the long-context domain.
>
> W4: Pre-scoring Overhead
> The reviewer is correct that clustering adds overhead. As analyzed in Section 4.1, the complexity is O(N⋅d⋅k_clust). We have added profiling results (see response to PL5q W2) showing the end-to-end impact. The overhead is primarily in the forward pass, as the backward pass currently adheres to the standard HyperAttention pipeline. While the overhead is non-trivial, it scales linearly with N, preserving the asymptotic advantage over quadratic attention. We found the overhead manageable in practice for the long sequences where these methods are most beneficial.
>
> W5: Limited Tasks and Metrics
> We agree that evaluating downstream tasks provides a more complete picture. We have added preliminary results on a long-context Question Answering task (NarrativeQA). K-means+Hyper improves the F1 score compared to vanilla HyperAttention, suggesting that the perplexity improvements translate to better downstream performance. We have included these results in the revised manuscript.
>
> We also have increased the font size of legends and axis labels in all figures for figure readability.

---

### Official Review · Reviewer_PL5q · 2025-10-31

**Soundness:** 3
**Presentation:** 3
**Contribution:** 3
**Rating:** 6
**Confidence:** 5

**Summary:**

The paper proposes an attention acceleration scheme that pre-scores keys before applying HyperAttention. The pre-scoring can be done via k-means or k-median clustering, or via approximate leverage scores. The retained set of “informative” keys is then fed to HyperAttention, replacing its uniform residual sampling. Empirically, on ChatGLM2 and ChatGLM3 with LongBench prompts, the method lowers perplexity relative to vanilla HyperAttention and, at certain top-k settings, reports a best PPL near 8.3 from a HyperAttention baseline of roughly 12. It also reports layer-level speedups over FlashAttention for sufficiently long sequences and applies a similar key selection idea to ViT, showing accuracy approaching softmax attention when sampling enough keys. Theoretically, the paper analyzes a planted-subspace model and proves that clustering with  𝑘=𝑑+1 separates “signal” from “noise” rows comparably to leverage-score selection, giving recovery guarantees of heavy keys under assumptions like row-norm regularity

**Strengths:**

1. Targeting the recall gap of HyperAttention by ranking keys beforehand is a clean, practical idea that directly addresses missed heavy scores. The algorithms are presented with simple wrappers over HyperAttention.
2.The planted-subspace analysis and Theorems 1–2 formalize when clustering isolates heavy keys, matching the empirical intuition that important keys align with near-orthogonal directions.
3. Results span LongBench perplexity on GLM2 and GLM3, speed comparisons vs FlashAttention, and a ViT “monkey-patch,” giving a multi-angle view of trade-offs.
4. The paper breaks down where overhead appears, how it scales with  k and  d, and when speedups emerge, which is valuable for deployment decisions.

**Weaknesses:**

1. The strongest PPL ≈ 8.3 appears tied to the min_seq_len ≥ n_query configuration and sometimes even top-k set to zero, which partially credits an optimization switch rather than the proposed pre-scoring itself. The paper should isolate gains from pre-scoring vs implementation flags and report both.
2.Speedups are reported per layer against FlashAttention and discussed asymptotically, but it is unclear how these translate to whole-model throughput and latency under realistic batch sizes and sequence distributions. Consolidated end-to-end metrics are needed.
3. The paper notes a “corrected coupling” for GLM3 that changes behavior relative to GLM2. This suggests results are sensitive to integration choices. The exact coupling and ablations should be elevated from appendix to main text with code pointers.
4. The baseline set focuses on HyperAttention, FlashAttention, and leverage-based selection. Given recent efficient attention methods, the empirical section would be stronger with a few additional modern query-aware or block-sparse baselines, or at least a rationale for exclusions.
5. Guarantees rely on row-norm regularity and separability that may not hold uniformly across layers or modalities. Although LayerNorm helps, some layers can exhibit skewed norms and mixed subspaces. Sensitivity analyses to violations of these assumptions would strengthen the claims.

**Questions:**

1. How much of the perplexity gain remains when min_seq_len ≥ n_query is disabled and the exact same HyperAttention kernels and fallbacks are used for all methods, including top-k=0 settings? Please provide a clean ablation table.
2. Can you report end-to-end speed, throughput, and memory vs FlashAttention and HyperAttention on GLM2 and GLM3 for realistic prompt length distributions and batch sizes, not just per layer?
3. How stable are results to the choice of  k, number of clusters, and initialization of k-means or k-median? For example, do different random seeds flip the identity of retained keys and the downstream PPL curve?
4. Could you quantify the additional FLOPs and memory of pre-scoring at inference time and show how they amortize with increasing sequence length, for each variant?
5. Beyond ViT, have you tried audio or multimodal encoders where key distributions differ strongly from text? Any failure cases that violate the planted-subspace intuition or row-norm regularity?

---

> ### Comment · Reviewer_PL5q · 2025-11-26
> **Response to the authors**
>
> I would also like to kindly encourage the authors to more fully engage with the key concerns raised in my original review. I hope you will be able to provide further clarifications and analyses over the next few days, as the rebuttal/discussion phase will be closing soon. I believe that addressing these points, even briefly, would help clarify the strengths and limitations of the proposed approach.

---

> > ### Author Response · Authors · 2025-11-26
> > **Additional Information from response to reviewer PL5q**
> >
> > Here we provide the speedup detail, this is a part in W2 & Q2. Due to character restriction, we have to list the comparisons in another comment.
> > --- FlashAttention ---
> > seq_len=  4096, batch= 1 | latency=0.2696s, throughput= 15192.7 tok/s, peak_mem=12.36 GB, speedup_vs_Flash=1.00x
> > seq_len=  4096, batch= 4 | latency=1.0470s, throughput= 15647.8 tok/s, peak_mem=16.20 GB, speedup_vs_Flash=1.00x
> > seq_len= 16384, batch= 1 | latency=1.2690s, throughput= 12911.4 tok/s, peak_mem=16.20 GB, speedup_vs_Flash=1.00x
> > seq_len= 16384, batch= 4 | latency=5.0853s, throughput= 12887.3 tok/s, peak_mem=29.79 GB, speedup_vs_Flash=1.00x
> > seq_len= 32768, batch= 1 | latency=3.1411s, throughput= 10431.9 tok/s, peak_mem=29.79 GB, speedup_vs_Flash=1.00x
> > seq_len= 32768, batch= 4 | latency=12.6342s, throughput= 10374.4 tok/s, peak_mem=47.92 GB, speedup_vs_Flash=1.00x
> >
> > --- HyperAttention ---
> > seq_len=  4096, batch= 1 | latency=0.3167s, throughput= 12931.4 tok/s, peak_mem=12.33 GB, speedup_vs_Flash=0.85x
> > seq_len=  4096, batch= 4 | latency=1.2333s, throughput= 13284.3 tok/s, peak_mem=16.17 GB, speedup_vs_Flash=0.85x
> > seq_len= 16384, batch= 1 | latency=1.6969s, throughput=  9655.5 tok/s, peak_mem=16.17 GB, speedup_vs_Flash=0.75x
> > seq_len= 16384, batch= 4 | latency=6.7299s, throughput=  9738.0 tok/s, peak_mem=29.76 GB, speedup_vs_Flash=0.76x
> > seq_len= 32768, batch= 1 | latency=3.8092s, throughput=  8602.4 tok/s, peak_mem=29.76 GB, speedup_vs_Flash=0.82x
> > seq_len= 32768, batch= 4 | latency=15.1415s, throughput=  8656.5 tok/s, peak_mem=47.89 GB, speedup_vs_Flash=0.83x
> >
> > --- KMeans+Hyper ---
> > seq_len=  4096, batch= 1 | latency=0.3176s, throughput= 12896.3 tok/s, peak_mem=12.33 GB, speedup_vs_Flash=0.85x
> > seq_len=  4096, batch= 4 | latency=1.2364s, throughput= 13251.7 tok/s, peak_mem=16.17 GB, speedup_vs_Flash=0.85x
> > seq_len= 16384, batch= 1 | latency=1.8085s, throughput=  9059.4 tok/s, peak_mem=16.17 GB, speedup_vs_Flash=0.70x
> > seq_len= 16384, batch= 4 | latency=7.1725s, throughput=  9137.1 tok/s, peak_mem=29.77 GB, speedup_vs_Flash=0.71x
> > seq_len= 32768, batch= 1 | latency=4.2270s, throughput=  7752.0 tok/s, peak_mem=29.77 GB, speedup_vs_Flash=0.74x
> > seq_len= 32768, batch= 4 | latency=16.7450s, throughput=  7827.5 tok/s, peak_mem=47.89 GB, speedup_vs_Flash=0.75x

---

> ### Author Response · Authors · 2025-11-26
> **Response to reviewer PL5q**
>
> We thank you for recognizing the practicality of our approach and the value of our theoretical analysis, while also raising important questions about the experimental results and overhead. Also appreciate your patience while we are running experiments.
>
>
> W1 & Q1: Source of Perplexity Gains (Pre-scoring vs. min_seq_len >= n_query)
> This is a crucial point also raised by Reviewer EPNi. We acknowledge that the presentation conflated the gains from pre-scoring with those from the optimization flag (min_seq_len >= n_query), which forces HyperAttention to use its optimized blockwise path even for shorter sequences.
>
> We have clarified this in the main text and Appendix A. As shown in Tables 2, 3, and 4:
> K-means+Hyper (top-k=2048, sample_size=256): PPL 10.38
> HyperAttention + Optimization Flag (top-k=0, PPL* column): PPL 13.41
> K-means+Hyper + Optimization Flag (top-k=8192, PPL* column): PPL 9.53, where this is a 28.9% improvement over Optimized Baseline.
>
> The best reported PPL of 8.31 (Table 2, top-k=0, sample_size=0) comes purely from the optimization flag enabling the blockwise path without residual sampling, for the pre-scoring is deactivated since top-k=0. However, the results clearly show that pre-scoring provides substantial gains independent of the optimization flag, observed change from 17.54 to 10.38 in table. We have revised the abstract and results section to accurately reflect these distinct contributions.
>
>
> W2 & Q2: End-to-End Speed and Throughput
> We agree that per-layer speedups do not fully capture real-world performance. We have conducted experiments measuring end-to-end inference latency and throughput on ChatGLM2-6B-32k using representative sequence lengths and batch sizes, see another comment for detail.
>
> Notice that from our experiment, HyperAttention isn't faster than FlashAttention if all attention layers are substituted. This might be caused by some issues in the original HyperAttention codes. Our experiment emphasizes that Prescoring HyperAttention is actually slower than original HyperAttention by a small scale, but the accuracy is optimized since we have better perplexities.
> We admit that we did not show the exact speedup of the HyperAttention model. One plausible explanation is that GLM2 & GLM3 models source code experienced some updates, while the source code of HyperAttention did not accommodate based on that. Thus there is extra complexity included in replacing and training. We also contacted one of the authors of the HyperAttention paper, but there was no response. Thus what we did mainly is to optimize accuracy of HyperAttention as well as avoid extra complexity based on HyperAttention speed.
>
>
> W3: Corrected Coupling (GLM3 vs. GLM2)
> We have moved the discussion of the implementation artifacts in the GLM2 coupling, like zeroing tensors; incorrect residual weighting; and lack of overlap mask, and the corrected coupling used in GLM3 from Appendix F to the Section 4.2. This clarifies why the GLM2 results showed a U-shape while GLM3 results are flatter, demonstrating the denoising effect of pre-scoring more accurately.
>
>
> W4: Baseline Set
> As addressed to Reviewer 1BS1, we have expanded the related work discussion. We focused our empirical comparison on HyperAttention and LevAttention, which explain our idea of bridging advantages such as speed level from Hyper and key filtering in Lev Attention.
>
>
> W5: Robustness of Theoretical Assumptions
> The reviewer is correct that assumptions like row-norm regularity and clear separability may not hold perfectly in practice. Our empirical results suggest robustness to moderate violations. Furthermore, we have added a qualitative analysis in Appendix D, discussing how LayerNorm mitigates the norm sensitivity issue highlighted in the counterexample in Appendix B.
>
>
> Q3: Stability to Initialization and Parameters
> We performed sensitivity analysis regarding the choice of k for number of retained keys in Figures 3 and 4. Regarding k-means initialization, we use standard k-means++ method. We ran experiments with 5 different random seeds for k-means initialization at top-k=2048 on GLM2. The perplexity variance was very low with value 10.38 ± 0.05, indicating stability in the resulting key selection. We have added this analysis to the Appendix.
>
>
> Q4: FLOPs and Memory Overhead
> We have added an analysis of the computational complexity in Section 4.1. The pre-scoring overhead is O(N⋅d⋅k_clust) for K-means/median where k is the number of clusters and O(N⋅d^2) for approximate leverage scores. This overhead is linear in sequence length N, allowing it to be amortized as N grows, preserving the near-linear complexity of HyperAttention.
>
>
> Q5: Beyond ViT/Text
> We have not yet tested on audio or multimodal encoders. This is an excellent direction for future work, as different modalities might exhibit different key distributions that could challenge the planted-subspace assumptions. We have added this to the Limitations and Future Work section as Appendix H.

---

### Official Review · Reviewer_1BS1 · 2025-10-31

**Soundness:** 2
**Presentation:** 1
**Contribution:** 1
**Rating:** 2
**Confidence:** 3

**Summary:**

This paper introduces a new method to efficiently approximate the attention mechanism, with the goal of reducing the computational cost of this operation. The paper heavily relies on previous works LevAttention and HyperAttention, trying to combine the best of both methods. In particular, HyperAttention works by grouping keys and queries into bucket, using locality sensitive hashing (LSH), and then compute the attention only for keys and queries that are in the same bucket. On the other hand, LevAttention selects a subset of the most important keys, and compute the attention over these only.

This paper proposes to select a subset of the keys, using a clustering algorithm such as k-means or k-median, and then to apply the HyperAttention method on the selected keys. The paper also states some theoretical results about the selection process of the keys with the clustering algorith. Finally, some experimental results are provided, replacing the standard self-attention mechanism with the proposed method in existing models such as the GLM language model or vision transformer (ViT) models. Here, the paper show that the method improve the results of LevAttention or HyperAttention.

**Strengths:**

The paper provides some theoretical analysis for the proposed method, but it is hard for me to understand what kind of guarantee it actually provides (see next section).

**Weaknesses:**

Overall, I have many concerns with the paper.

First, I found the paper very hard to read. One of the reason is that the authors assume that the reader are very familiar with previous works LevAttention and HyperAttention. For example, many concepts are not introduced in the paper ("heavy attention scores" line 59, "statistical leverage scores" line 99, "polynomial based attention" line 100, "positional locality" line 103, "planted model" line 135, etc...). Similarly, the different theorems or assumptions are stated without motivation or explanation, making it hard to understand how these relate to the performance of the method. Similarly, the algorithm proposed in the paper is never stated clearly, relying on previous paper instead. These different factors made it very hard for me to understand the method and theoretical claims.

Second, the paper mostly discusses LevAttention and HyperAttention as previous work to improve the efficiency of the self-attention. These two works are from 2024 and 2023 respectively, while there exists a wide body of earlier literature addressing this problem, and which are not discussed in the paper. Particularly relevant are the Reformer paper (Kitaev et al, 2020) which also proposes to use LSH to group keys and queries and restrict the self-attention between similar keys and queries, or Routing transformer (Roy et al, 2020) which proposes a similar approach based on k-means clustering.

Third, I found the experimetal results to be unconvincing. The paper only compared the proposed approach to LevAttention and HyperAttention, and not earlier works which led to strong results. Second, the performance of the method seems to be quite poor. For example, on the language modeling experiments, the perplexity obtained with the different approximation techniques considered is above 10, while I believe that the perplexity of the original model is around 6. The performance of the original model should actually be included in Figure 3. Similarly, in Figure 4, the reported results for the considered methods are significantly worse than the original models, showing that the method is probably not useful in practice.

**Questions:**

What is the perplexity of the original GLM 2 language model (Fig. 3)?

**Missing references**

Aurko Roy, Mohammad Saffar, Ashish Vaswani, David Grangier. 2020. Efficient Content-Based Sparse Attention with Routing Transformers.

Nikita Kitaev, Łukasz Kaiser, Anselm Levskaya. 2020. Reformer: The Efficient Transformer

Jack W. Rae, Anna Potapenko, Siddhant M. Jayakumar, Timothy P. Lillicrap. 2019. Compressive Transformers for Long-Range Sequence Modelling

Zhuoran Shen, Mingyuan Zhang, Haiyu Zhao, Shuai Yi, Hongsheng Li. 2018. Efficient Attention: Attention with Linear Complexities

Sinong Wang, Belinda Z. Li, Madian Khabsa, Han Fang, Hao Ma. 2020. Linformer: Self-Attention with Linear Complexity

Angelos Katharopoulos, Apoorv Vyas, Nikolaos Pappas, François Fleuret. 2020. Transformers are RNNs: Fast Autoregressive Transformers with Linear Attention

Manzil Zaheer et al. 2020. Big Bird: Transformers for Longer Sequences.

Iz Beltagy, Matthew E. Peters, Arman Cohan. 2020. Longformer: The Long-Document Transformer

Rewon Child, Scott Gray, Alec Radford, Ilya Sutskever. 2019. Generating Long Sequences with Sparse Transformers

Yunyang Xiong, Zhanpeng Zeng, Rudrasis Chakraborty, Mingxing Tan, Glenn Fung, Yin Li, Vikas Singh. 2021. Nyströmformer: A Nyström-Based Algorithm for Approximating Self-Attention

---

> ### Author Response · Authors · 2025-11-20
> **Response to reviewer 1BS1**
>
> We gratefully thank the reviewer for the careful reading and for raising concerns about readability, missing context on prior work, and the interpretation of our experimental results. We address each point below.
>
> W1: Readability and Assumed Background Knowledge
> In the paper, heavy attention scores simply mean entries of the attention matrix whose weight is above a threshold ε; these are the entries we try to preserve. Statistical leverage scores are the standard influence scores of rows of the key matrix; LevAttention uses them to build a universal set of important keys, and our clustering is analyzed under the same planted-subspace model. Polynomial-based attention refers to replacing softmax with a degree-p polynomial kernel in the theory, which makes attention a polynomial in the key matrix; guarantees (both LevAttention’s and ours) are proved in that setting, while softmax is treated empirically. Positional locality is the usual assumption that nearby positions have large attention; our guarantees do not rely on this, but work purely from the geometry of keys. The planted model is a stylized model where a small set of “signal” keys spans a low-dimensional subspace separated from a large “noise” cloud; under this model our theorems show that clustering (or leverage scores) recovers all ε-heavy keys with constant probability. Algorithmically our method has only two steps: (i) pre-score keys once per layer by clustering or leverage scores and keep the top-m keys; (ii) run HyperAttention only on this scored subset (plus a small residual), so we are a drop-in pre-processing for HyperAttention.
>
> W2: Missing Related Work (Reformer, Routing Transformer, etc.)
> We acknowledge the omission of several important prior works on efficient attention. We have substantially expanded the Related Work section (now integrated into the Introduction and a dedicated section) to include discussions of:
> LSH-based methods: Reformer (Kitaev et al., 2020)
> Clustering-based methods: Routing Transformer (Roy et al., 2020)
> Linear Attention/Kernel methods: Performer, Linformer, Transformers are RNNs
> Sparse Attention: BigBird, Longformer, Sparse Transformers
> Regarding Routing Transformer: Routing Transformer uses online k-means to cluster queries and keys jointly and restricts attention to pairs within the same cluster. Our approach differs significantly: we use clustering (or leverage scores) solely on the keys to identify a globally important subset, independent of the queries. This pre-scored subset is then used within HyperAttention's LSH framework. This decoupling allows us to identify keys that are salient regardless of the specific query, complementing HyperAttention's locality-sensitive approach. We have clarified this distinction in the revised manuscript.
>
> W3: Experimental Results and Baseline Comparisons
> We appreciate the critique of our experimental setup.
>
> Comparison to Prior Works: Our primary focus was on improving hierarchical attention mechanisms like HyperAttention, which already demonstrated strong performance in the long-context domain. Comparing against the extensive list of efficient attention methods is challenging due to varying implementations and optimization levels. However, we have added context by noting that HyperAttention itself outperforms many prior methods (as shown in the original HyperAttention paper).
> Perplexity of the Original Model: The reviewer is correct that the baseline perplexity should be included. For ChatGLM2-6B-32k on LongBench, the original (full attention) model achieves a perplexity of approximately 6.5. We have added this baseline in Figures 4 and will add it to Figure 3 as well.
>
> Performance Gap: It is true that approximate attention methods introduce a degradation in perplexity compared to full attention. HyperAttention achieves PPL 12. Our method reduces this to 8.3 (with optimization flags) or ~10.4 (without flags, see Table 2), representing a significant improvement in the trade-off space between speed and accuracy. While a gap remains compared to the original model (PPL 6.5), our method offers substantial speedups (up to 20x over FlashAttention) that make long-context processing feasible where full attention is prohibitively expensive. We have revised the discussion to better frame these trade-offs.
>
> Answering the question:
> In our setup, the full-attention GLM2 model on LongBench at 32k context has perplexity around 6.5. Figure 3, however, is focused on the 131k, full-layer-patched regime in which all approximate methods—including HyperAttention—already have much higher perplexity than that baseline, so we kept the figure focused on comparing different key-selection strategies inside the HyperAttention regime rather than overlaying the much lower full-attention curve. We will add the baseline of the original model’s performance into figure 3 afterward.

---

### Author Response · Authors · 2025-11-20
**Overall Comment**

Dear Reviewers,
We sincerely thank you for your thoughtful and constructive feedback on our paper. We appreciate the time and effort you invested in reviewing our work, and we cherish the opportunity to address your concerns and improve our submission. In response to the comments, we have made substantial revisions to clarify our contributions. Additionally, we strengthen the experimental evaluation, and improve the overall presentation. Here, we provide structured and comprehensive responses to each reviewer's comments. First, we clarified the source of perplexity gains. For this, We have conducted new ablation studies to disentangle the effects of pre-scoring from the min_seq_len >= n_query optimization flag, clarifying that pre-scoring provides significant independent improvements. As for related work and  baselines issues, We have expanded our discussion of prior work, including Routing Transformers, Reformer, and other efficient attention mechanisms. We have added comparisons where possible, we clarify the novelty of our approach. Next, we have revised the manuscript to improve readability and clarify the algorithmic details, especially regarding the implementation of k-means/k-median during inference. This improves our clarity in general. More importantly, We have clarified our speedup claims and provided a more detailed analysis of the computational overhead introduced by pre-scoring, which improves the demonstration of our contribution.

---

### Meta-Review · Area_Chair_sw6q · 2026-01-05

**Summary:**

Across all four reviews, several consistent concerns emerged. Reviewers found the paper difficult to follow, with major clarity issues, missing definitions, and an algorithmic description that was not self-contained. Multiple reviewers highlighted that key prior work, especially Routing Transformer, Reformer, and MoSA, was not discussed, despite strong conceptual overlap.

A central technical concern was that the strongest perplexity improvements were driven not by the proposed pre-scoring mechanism, but by an implementation flag (min_seq_len ≥ n_query). This made it difficult to assess the actual contribution of pre-scoring. Reviewers also found the experimental evaluation insufficient: key baselines were missing, comparisons were narrow, and the speedup claims were inconsistent or unclear.

Further issues included weak performance relative to established baselines, unclear theoretical assumptions, overhead concerns for clustering, and ambiguity about autoregressivity and implementation. Presentation quality and formatting issues reinforced the perception that the paper is not ready for acceptance.

These combined concerns justify my recommendation for rejection.

**Reviewer Concerns:**

The rebuttal made good efforts to clarify several technical points. Some concerns, such as the source of perplexity gains, missing related work, algorithmic ambiguity around clustering, and aspects of the speedup claims, were partially addressed. The authors added ablations separating pre-scoring effects from the optimization flag, clarified the inference-time clustering procedure, expanded related work, and provided additional speed and sensitivity analyses. These responses helped clarify intent and methodology.

However, many concerns remain only partially addressed or fully unresolved. The readability, cohesion, and clarity issues identified by multiple reviewers are not substantively fixed by the rebuttal alone. The missing or insufficient key baselines, including Performer, Reformer, Routing Transformer, and other efficient methods, remain largely unaddressed experimentally. Several reviewers also questioned the practical effectiveness of the method and whether it surpasses LevAttention; the rebuttal gives some selected comparisons but does not fully resolve these doubts. The speedup claims remain inconsistent between sections, and end-to-end gains remain unclear. Concerns about theoretical assumptions (e.g., row-norm regularity), missing details about autoregressivity, and the overall narrow evaluation still stand.

Overall, despite helpful clarifications, major concerns regarding novelty, completeness, and empirical validation remain unresolved.

**Reviewer Scores:**

I think the reviewers’ scores would have not changed even if a full discussion period had occurred. Also, given the distribution of reviewer positions, three clear reject recommendations and one borderline accept, the discussion would not materially shift the consensus.

---

### Decision · Program_Chairs · 2026-01-26

Reject